# *Bifidobacterium* Relieved Fluoride-Induced Hepatic and Ileal Toxicity via Inflammatory Response and Bile Acid Transporters in Mice

**DOI:** 10.3390/foods13071011

**Published:** 2024-03-26

**Authors:** Yue Wu, Ao Cheng, Yu Wang, Qianlong Zhu, Xuting Ren, Yiguang Lu, Erbao Shi, Cuicui Zhuang, Jundong Wang, Chen Liang, Jianhai Zhang

**Affiliations:** 1College of Veterinary Medicine, Shanxi Agricultural University, 1 Mingxian South Road, Taigu 030801, China; 2College of Veterinary Medicine, Inner Mongolia Agricultural University, Huhehot 010018, China; 3College of Animal Science, Shanxi Agricultural University, Taigu 030801, China

**Keywords:** intestinal microbiota, NaF, hepatointestinal injury, inflammatory factors, bile acid

## Abstract

Fluoride is a pervasive environmental contaminant. Prolonged excessive fluoride intake can inflict severe damage on the liver and intestines. Previous 16S rDNA sequencing revealed a decrease in ileal *Bifidobacterium* abundance during fluoride-induced hepatointestinal injury. Hence, this work aimed to investigate the possible mitigating function of *Bifidobacterium* on hepatointestinal injury caused by fluoride. Thirty-six 6-week-old C57BL/6J mice (equally divided between males and females) were allotted randomly to three groups: Ctrl group (distilled water), NaF group, and NaF + Ba group (100 mg/L NaF distilled water). After 10 weeks, the mice were given 1 × 10^9^ CFU/mL *Bifidobacterium* solution (0.2 mL/day) intragastrically in the NaF + Ba group for 8 weeks, and the mice in other groups were given the same amount of distilled water. Dental damage, bone fluoride content, blood routine, liver and intestinal microstructure and function, inflammatory factors, and regulatory cholic acid transporters were examined. Our results showed that fluoride increased glutamic-oxalacetic transaminase (GOT), glutamic-pyruvic transaminase (GPT) activities, and the levels of lipopolysaccharide (LPS), IL-1β, IL-6, TNF-α, and IL-10 levels in serum, liver, and ileum. However, *Bifidobacterium* intervention alleviated fluoride-induced changes in the above indicators. In addition, *Bifidobacterium* reduced the mRNA expression levels of bile acid transporters ASBT, IBABP, OST-α, and OST-β in the ileum. In summary, *Bifidobacterium* supplementation relieved fluoride-induced hepatic and ileal toxicity via an inflammatory response and bile acid transporters in the liver and ileum of mice.

## 1. Introduction

Fluoride is a pervasive environmental contaminant, primarily found in groundwater [1]. The fluoride concentration in the drinking water of over 180 million people exceeds the thresholds recommended by the World Health Organization, leading to global public health concerns [2]. Research indicates that chronic fluoride exposure results in severe hepatocellular vacuolation, and damage to the rough endoplasmic reticulum and mitochondria of hepatocytes, affecting liver metabolism, diminishing hepatic detoxification capacity, and causing varying degrees of injury [3,4,5]. Elevated fluoride intake may lead to intestinal mucosal damage, triggering inflammatory responses in the intestine and impacting intestinal microbiota, resulting in gut dysbiosis [6,7,8]. Prolonged excessive fluoride intake can inflict severe damage on the liver and intestines. However, its underlying mechanisms remain unclear.

The intestinal microbiota plays a pivotal role in maintaining the integrity of the intestinal barrier, and its diversity is intricately linked to the overall health of the host [9]. Dysbiosis of the intestinal microbiota can precipitate various health ailments [10]. In recent years, research has revealed a close association between environmental fluoride exposure and intestinal inflammation and microbiota dysbiosis [11,12]. Fluoride substantially alters the components of the mouse intestinal microbiota [13,14]. Similarly, another study reported that sodium fluoride (NaF) disrupts the balance of the intestinal microbiota, potentially causing intestinal dysfunction in mice [15]. It is noteworthy that dysfunction and ecological imbalance of the intestinal barrier can also lead to liver damage [16]. In our previous study, based on 16S rDNA sequencing, fluoride exposure markedly altered the components of the mouse ileal microbiota, particularly reducing the abundance of the *Bifidobacterium* genus.

*Bifidobacterium*, as the predominant microbiota in the mammals and poultry host’s gastrointestinal tract, facilitates digestion, enhances nutrient absorption, boosts immune function, and preserves intestinal flora balance, thereby playing an indispensable role in overall health maintenance [17]. A recent study discovered *Bifidobacterium* can regulate bile acid metabolism through its bile acid hydrolase activity and alleviate the liver and intestinal damage caused by many factors (high-fat diet, non-steroidal anti-inflammatory drugs, harmful bacteria, etc.) [18]. Adhesin secreted by *Bifidobacterium* can significantly alleviate the inflammatory reaction of intestinal epithelial cells induced by LPS and H_2_O_2_, and protect the intestinal tract [19]. Current studies have observed that *Bifidobacterium* mitigated liver inflammation by altering the intestinal flora, thereby ameliorating non-alcoholic fatty liver disease, notably by diminishing the levels of LPS and inflammatory mediators in both serum and liver [20,21]. Therefore, the role of *Bifidobacterium* in fluoride-induced hepato-intestinal toxicity urgently needs further investigation.

This study established a *Bifidobacterium*-supplemented mouse model subjected to fluoride treatment. Subsequently, fluoride ion concentration in the femur, serum indicators of liver function (GOT and GPT), inflammatory factor (IL-1β, IL-6, IL-10, and TNF-α) content, and mRNA expression of genes associated with bile acid transport receptors in the liver (NTCP, MRP2, and BSEP) and ileum (ASBT, IBABP, and OST-α/β) were further detected. The research aimed to investigate *Bifidobacterium*’s protective effect against fluoride-induced hepatointestinal injury, focusing on inflammation. Additionally, it sought to elucidate the correlation between hepatointestinal injury resulting from environmental fluoride exposure and *Bifidobacterium* in the ileum and provide a reference for the influence of environmental factors on the imbalance of intestinal flora.

## 2. Material and Methods

### 2.1. Animals and Treatments

Male and female C57BL/6J mice, aged 6 weeks, were obtained from China Eaton Leverage Company Limited and bred in the laboratory. Eighteen mice of each gender were randomly assigned to three groups: control, sodium fluoride (NaF), and sodium fluoride with *Bifidobacterium* intervention (NaF + Ba). Each group comprised 12 mice, with an equal gender distribution. After the 10th week, the *Bifidobacterium* intervention began. Each NaF + Ba group of mice was orally administered 0.2 mL of *Bifidobacterium* solution per day, while the remaining groups were given the same amount of distilled water for 8 weeks. Subsequently, the mice were slaughtered, and the necessary samples were collected.

The mice were housed under controlled conditions with a temperature range of 20–25 °C and humidity between 50–60%, ensuring adequate ventilation and hygiene standards. They were provided ad libitum access to water and food (obtained from Jiangsu Collaborative Pharmaceutical and Biological Engineering Company Limited (Nanjing, China)). The feeding regimen spanned 18 weeks, during which weekly weight measurements were conducted. Sodium fluoride (NaF) was procured from Tianjin Chemical Reagent Factory No.3 (Tianjin, China), while *Bifidobacterium* JYBR-190 lyophilized powder was sourced from Shandong Zhongke Jiayi Biological Engineering Company Limited (Weifang, China). All animal procedures strictly adhered to the regulations and guidelines outlined by the Laboratory Animal Ethics Committee of Shanxi Agricultural University.

### 2.2. Organ Coefficient Analysis

The body weight of all mice was recorded before sacrifice, as was the weight of each organ weight after dissection. The organ coefficient was calculated using the formula: organ coefficient ratio = organ wet weight (mg)/body weight (g).

### 2.3. Bone Fluoride Determination

The fluoride content in the femur was assessed using the fluoride ion-selective electrode method. Following slaughter, both right and left femurs were removed, and the residual soft tissue on the surface was cleaned with medical gauze. Subsequently, the samples underwent drying at 105 °C for 4 h, followed by ashing in a muffle furnace at 550 °C for 4 h. Bone ash samples weighing 20–25 mg were dissolved in 0.25 mol/L hydrochloric acid; the pH was adjusted to 4 using sodium hydroxide, and the volume was made constant with buffer solution. Finally, the fluoride content was determined using a fluoride ion-selective electrode [22].

### 2.4. Grading of Dental Damage

The evaluation of fluoride-induced dental damage was conducted using an 11-point grading scale [23]. The criteria for judging included transparency of the enamel mandibular incisors, color and size of chalky plaques, and degree of tooth defects. On the whole, grade 0 was pigeonholed as normal, grades 1–3 were classified as very mild dental damage, grades 4–5 were considered as mild dental damage, grades 6–8 were rated as moderate dental damage, and grades 9–10 were categorized as severe dental damage.

### 2.5. Blood Routine

Blood samples were analyzed by routine blood analysis instruments, including the number of lymphocytes, middle cells (sum of monocytes, eosinophils, and basophils), neutrophils, red blood cells, white blood cells, hemoglobin, and platelets.

### 2.6. Histopathological Examination

Liver and intestinal tissues (including the duodenum, ileum, colon, and cecum) were fixed in a 10% formalin solution. Following a 24-h fixation period, the tissues were rinsed, embedded in paraffin, and sectioned at a thickness of 5 µm. The sections were then stained with haematoxylin and eosin (H&E) solution and examined using light microscopy.

### 2.7. Biochemical Assays

Liver enzymes GOT, GPT, and TP levels were determined using commercial detection kits (Nanjing Jiancheng Bioengineering Institute, Nanjing, China) utilizing a microplate reader (Thermo Fisher, Waltham, MA, USA) following the manufacturer’s protocols.

### 2.8. ELISA Analysis

We exploited ELISA to detect LPS and inflammatory factor levels in the serum, liver, and ileum. Ileum and liver tissues weighing about 20 mg were homogenized in 180 μL PBS on ice. The liver tissue homogenate, ileum tissue homogenate, and serum were then centrifuged at 4 °C, 3000 rpm for 30 min. Afterward, 10 μL of supernatant was diluted five times, and the levels of LPS, IL-1β, IL-6, IL-10, and TNF-α were measured using ELISA kits following the manufacturer’s instructions. All enzyme-linked immunosorbent kits were purchased from Shanghai Jianglai Biological Company (Shanghai, China).

### 2.9. Real-Time PCR

Following the manufacturer’s protocol, total RNA was separated from liver and ileum tissues using Trizol Reagent (Beijing Quanshijin Biotechnology Company Limited, Beijing, China). A NanoDrop 2000 spectrophotometer (Thermo Fisher, USA) was used to detect the concentration and quality of RNA. Subsequently, the GoScript Reverse Transcriptase Reverse Transcription kit (Plommeg Biotechnology Company Limited, Shanghai, China) was used to reverse transcript total RNA into cDNA. Quantitative real-time PCR (qRT-PCR) was conducted with the GoTaq ^®^ qPCR System (Beijing Quanshijin Biotechnology Company Limited, Beijing, China) using the MX3000P system (Stratagene, La Jolla, CA, USA). Primers (Table 1) designed online by Primer 3.0 plus were synthesized by China Shanghai Sangon Biotech Company Limited. The genes of interest included bile acid salt export protein (BSEP), apical sodium-dependent bile acid transporter (ASBT), organic solute transporters alpha and beta-α (OST-α), and organic solute transporters alpha and beta-β (OST-β). The thermocycling conditions were as follows: initial denaturation at 94 °C for 5 min, followed by 40 amplification cycles at 94 °C for 15s, 60 °C for 30s, and 72 °C for 15s. Finally, a melting curve analysis was performed at 95 °C for 15s, 94 °C for 60s, and 94 °C for 15s.

### 2.10. Statistical Analysis

All statistical analysis was carried out using GraphPad Prism 8 with a one-way analysis of variance with Tukey’s test. Data are displayed as means ± SEM. * means significant differences between the control group and the NaF group/the NaF + Ba group. ^#^ represents significant differences between the NaF group and the NaF + Ba group. Statistical significance is defined as a *p*-value below 0.001, 0.01, or 0.05.

## 3. Results

### 3.1. Mouse Model Design and Primary Evaluation of Fluoride Exposure and Bifidobacterium Intervention

The experimental design included recording and presenting the body weights of mice throughout the study, as depicted in Figure 1A–C. There was no statistically significant change in body weight, as well as the organ coefficients of the liver, kidney, spleen, left epididymis, testis, ovary, uterus, and brain among mice (Table 2 and Table 3). This suggests neither NaF nor/or *Bifidobacterium* treatment affected growth in mice.

Incisors of mice were well developed, with enamel luster and normal transparency in the Ctrl group. In the NaF group, the incisors became shorter, with chalky in a large area and severe cutting ends, showing 100% moderate dental damage symptoms. In the NaF + Ba group, the proportions of dental damage were 1/3 moderate dental damage and 2/3 middle dental damage, respectively. Otherwise, there was abrasion at the end of the incisors, and white spots and little pigmentation scattered in the middle of them (Figure 1D,E). In the same period, compared to the Ctrl group, bone F^−^ concentrations were increased significantly in both the NaF group and NaF + Ba group (*p* < 0.001), but no significant change was observed among the experimental groups (Figure 1F). These results suggest the fluoride-exposed mouse model was successfully established, and *Bifidobacterium* intervention could mitigate the grade of dental damage.

### 3.2. Bifidobacterium Alleviates Fluoride-Induced Liver and Ileal Inflammation

As seen from the routine blood indicators level in Figure 2, compared to the Ctrl group, the number of middle cells, neutrophils, and white blood cells were obviously increased in the NaF + Ba group (*p* < 0.05, *p* < 0.01), except there were no alterations in lymphocytes, red blood cells, hemoglobin concentration, and platelets, and no pronounced difference were noticed in the NaF group. Only lymphocytes, middle cells, neutrophils, and white blood cell numbers in the NaF + Ba group were higher than in the NaF group (*p* < 0.05), and other indicators had no notable changes. The results suggest that *Bifidobacterium* affects the immune response in mice.

The expression of inflammatory cytokines including pro-inflammatory factors IL-1β, IL-6, TNF-α, and anti-inflammatory factor IL-10 are shown in Figure 3. Compared to the Ctrl group, the level of serum, liver, and ileum inflammatory cytokines in the NaF group were significantly elevated after 10-weeks fluoride exposure (*p* < 0.05, *p* < 0.01, *p* < 0.001); In the NaF + Ba group, an obvious increment was only found in IL-10 and serum TNF-α after 8 weeks of *Bifidobacterium* intervention (*p* < 0.05, *p* < 0.001). Compared to the NaF group, the pro-inflammatory factor consistency was obviously reduced except for no notable alteration in the level of serum TNF-α (*p* < 0.05, *p* < 0.01, *p* < 0.001), and a dramatically enhanced anti-inflammatory factor density was observed in the NaF + Ba group (*p* < 0.05). The above results indicate that *Bifidobacterium* alleviated the inflammation occurring in the liver and ileum of fluoride-treated mice.

### 3.3. Bifidobacterium Restored the Alteration of Morphology of Liver Induced by Fluoride

The morphological changes of the liver tissue were observed by H.E staining and are shown in Figure 4A. The hepatocytes were regularly arranged around the central vein with a round shape, clear cell boundary, and round or oval nuclei in the Ctrl group and NaF + Ba group, while the histopathological lesions, manifested as a disordered arrangement of hepatocytes, blurred boundary, and Kupffer cell number upregulation occurred in the liver tissue of the NaF group.

The biochemical indicators related to liver function were detected. In comparison with the Ctrl group, GOT and GPT activity significantly increased in the NaF group (*p* < 0.05) and NaF + Ba group (*p* < 0.01), without a significant difference in TP concentration (Figure 4B–D). All indexes demonstrated no dramatic changes in the NaF + Ba group in comparison to the NaF group. The results showed that *Bifidobacterium* alleviated fluoride-induced liver inflammation but not impaired liver function.

### 3.4. Bifidobacterium Intervention Alleviates Fluoride-Induced Intestinal Injury in Mice

The morphological characteristics of duodenum, ileum, colon, and cecum tissues in mice are shown in Figure 5A. In the Ctrl group, the intestinal epithelial cells had a neat arrangement and clear edges, intact intestinal mucosa, without significant injury. In the NaF group, the intestinal epithelial cells were disorderly arranged with a blurred edge of the nucleus, reduced height of the intestinal villi, damaged structure, velvet hollowing out, and rupture to varying degrees. The NaF + Ba group displayed mild intestinal injury with a slight reduction in villus length and disordered arrangement of intestinal epithelial cells.

The changes of LPS concentration in serum, ileum, and liver are shown in Figure 5B–D. In comparison to the Ctrl group, a significant increase was found in serum and ileum LPS concentrations (*p* < 0.05, *p* < 0.001) of the NaF group with no pronounced differences in the liver. The concentration of liver and ileum LPS in the NaF + Ba group had no notable differences except that serum LPS increased markedly (*p* < 0.001). Compared with the NaF group, the concentration of LPS was decreased significantly in serum (*p* < 0.05), with no obvious alteration in the liver and ileum of the NaF + Ba group. The results implied *Bifidobacterium* was able to restore the imbalance of homeostasis of intestinal barrier in mice caused by fluoride.

### 3.5. Bifidobacterium Regulated Bile Acid Transporters in Enterohepatic Circulation

In order to verify the changes in bile acid transporter after fluoride or/and *Bifidobacterium* treatment, the mRNA expression level of *ASBT*, *IBABP*, *OST-α*, and *OST-β* in the ileum, *NTCP*, *BSEP*, and *MRP2* in the liver were tested (Figure 6A–G). The distribution of bile acid transporters in the hepatic–intestinal circulation is shown in Figure 6H. In the ileum, compared to the Ctrl group, the mRNA expression of *ASBT*, *IBABP*, *OST-α*, and *OST-β* were notably decreased in the NaF + Ba group (*p* < 0.05, *p* < 0.01, *p* < 0.001), and there were not any notable changes in the NaF group. In parallel, all these genes decreased markedly between the NaF + Ba group and the NaF group (*p* < 0.05, *p* < 0.001). In the liver, no pronounced changes were present in the mRNA expressions of *NTCP*, *BSEP*, and *MRP2* among groups. The above results indicate that fluoride did not induce changes in bile acid transport in mice in this study. It is worth noting that *Bifidobacterium* may influence bile acid transporters in mice by reducing *ASBT*, *IBABP*, *OST-α*, and *OST-β* mRNA expression levels.

## 4. Discussion

Fluorosis can be categorized into different types, including drinking water-induced, tea-induced, coal-burning-induced, and industrial-induced fluorosis, with drinking water-induced fluorosis being the most significant due to contaminated water sources [24]. So, our research group chose to establish a fluoride-exposed mouse model by drinking water freely. The fluoride concentration in groundwater in China, India, and Pakistan has reached 48 mg/L [25], and the fluoride concentration in wastewater from copper smelting can exceed 10,000 mg/L [26]. In addition, the oral LD50/24 h of NaF for mice is 97.7 mg/kg body weight [27]. Based on the above background and previous fluoride studies in the laboratory [28], we constructed fluoride-treated mouse models by freely drinking distilled water containing 100 mg/L sodium fluoride for 18 weeks. Of note, *Bifidobacterium* was regarded as an effective therapeutic approach for alleviating liver and intestinal damage [29]. Here, the administration of 1 × 10^9^ CFU/mL *Bifidobacterium* is based on the research by Xin et al. [30].

The liver occupies an important role in toxicology studies. Most of the poisons are absorbed and metabolized through the liver [31]. Fluoride can accumulate in the liver and cause damage to the microstructure and function of the liver [3]. In this study, the normal morphological structure was destroyed post-fluoride exposure and mitigated after *Bifidobacterium* interposition. Both serum GPT and GOT activities were significantly increased in the NaF group. Here, *Bifidobacterium* did not alleviate the fluoride-induced impaired liver function.

Bones serve as the primary site for fluoride accumulation [32]. The study findings revealed that *Bifidobacterium* supplementation did not alleviate the accumulation of fluoride in bone. This indicates that *Bifidobacterium* has no effect on the absorption and excretion of fluoride ions, and *Bifidobacterium* may alleviate liver and intestine injury caused by fluoride in other ways. The results of the blood routine imply that we consider the role of the inflammatory response after *Bifidobacterium* supplementation. For that, a further study demonstrated that *Bifidobacterium* relieved the fluoride-induced increase in pro-inflammatory factors IL-1β, IL-6, and TNF-α in sera, liver, and ileum. *Bifidobacterium* can enhance the activity of natural killer cells and phagocytes, stimulate the activation, proliferation, and secretion of B lymphocytes, and boost the body’s immune response [33]. Our results indicated that *Bifidobacterium* can significantly increase the number of leukocytes, granulocytes, as well as intermediate cells (the sum of monocytes, eosinophils, and basophils) in blood. These results indicate that *Bifidobacterium* may alleviate liver and intestinal injury resulting from fluoride exposure by eliciting an anti-inflammatory response.

Fluoride is mainly absorbed and metabolized by gastrointestinal tissue, so the gastrointestinal tract is one of the tissues and organs most seriously affected by fluorosis [34]. Studies have found that patients with skeletal fluorosis have gastrointestinal symptoms after sampling and following tissue biopsy of the intestine, rupture of duodenal microvilli and abrasion on the mucosal surface were found [35]. The result stated clearly that *Bifidobacterium* intervention alleviated the fluoride-induced morphological structures of duodenum, ileum, colon, and cecum in mice. In our unpublished study, the ileum was the most sensitive to fluoride, and fluoride treatment significantly altered the microbial composition of ileal contents. The alteration in microbial composition prompted us to investigate the potential role of these microbes in inflammatory processes, particularly in the context of whether lipopolysaccharides (LPS) from Gram-negative cell walls exacerbate inflammation. LPS, a vital component of Gram-negative bacterial cell walls, releases inflammatory mediators through the activation of Kupffer cells in the liver, which in turn promote increased inflammation. In general, abnormal changes in LPS are also considered a marker of an imbalance of intestinal flora [36]. Nonetheless, the therapeutic role of *Bifidobacterium* in gastrointestinal diseases caused by fluoride is less studied. Former studies reported that when the intestine is stimulated for a long time, it can decrease the number of *Bifidobacterium*, destroy the integrity of the intestinal barrier, and release LPS entering the blood [37]. The present study discovered that *Bifidobacterium* can mitigate fluoride-induced damage to the intestinal microstructure and reduce the elevation of inflammatory factors in the ileum as well as LPS levels in the serum. These results suggest that *Bifidobacterium* restores the decline of intestinal barrier function resulting from fluoride exposure.

Our unpublished study has shown *Bifidobacterium* intervention significantly reduces the total bile acids concentration in serum and liver and alleviates the abnormalities of bile acid synthase and ileal FXR signaling pathway in fluoride-treated mice. Accordingly, we speculate *Bifidobacterium* could alleviate the hepatointestinal toxicity of fluoride through the hepatointestinal circulation metabolism of bile acid. For this, in this study, we further examined the bile acid transport receptors; however, fluoride did not cause the change of bile acid transporter, which did not confirm our previous conjecture. Interestingly, *Bifidobacterium* reduced bile acid receptor mRNA expression levels in the liver and ileum, and the specific mechanism remains to be further explored.

## 5. Conclusions

In conclusion, fluoride interfered with the enterohepatic circulation by altering the expression of serum liver function indicators GOT and GPT; pro-inflammatory factors IL-1β, IL-6, and TNF-α; anti-inflammatory factor IL-10; LPS; and intestinal barrier integrity, which then led to hepatointestinal injury. *Bifidobacterium* intervention ameliorated fluoride-induced damage to the liver and ileum in vivo, manifested as relief of inflammatory response and morphological structure (Figure 7). Additionally, *Bifidobacterium* intervention affected bile acid transport receptors. This study offers a theoretical basis and direction for the *Bifidobacterium* remission mechanism and reveals the connection between environmental fluoride exposure and *Bifidobacterium* in the gut.

## Figures and Tables

**Figure 1 foods-13-01011-f001:**
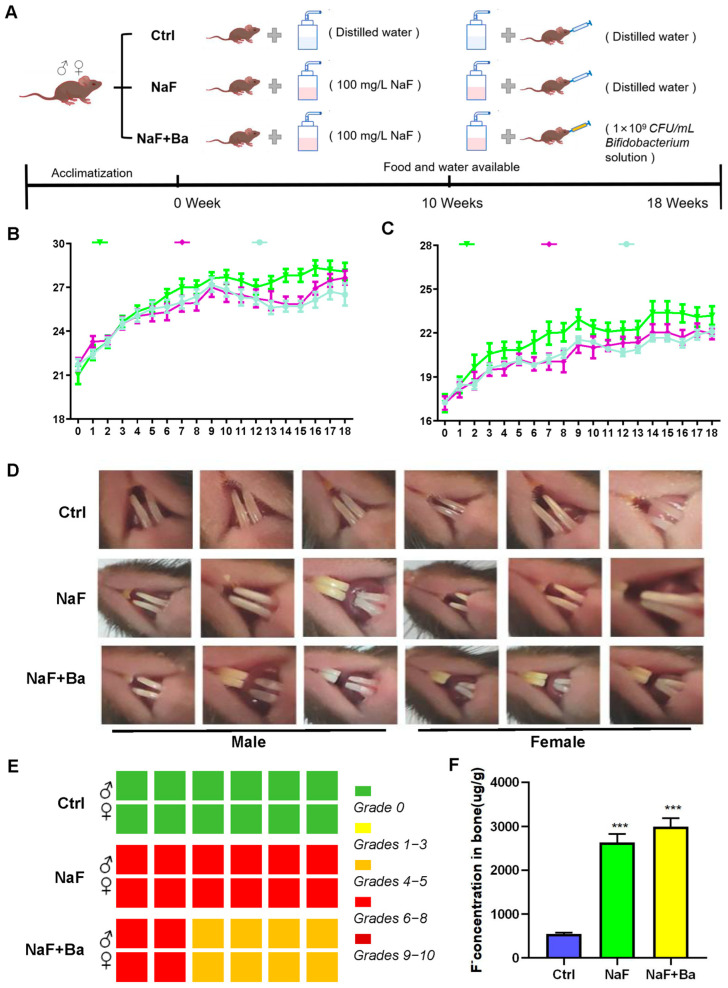
Fluoride exposure and *Bifidobacterium* intervention mouse model evaluation. (**A**) Schematic diagram of mouse model design and treatment. (**B**,**C**) Changes in body weight of mice from the Ctrl, NaF, and NaF + Ba groups. (**D**) Representative images of surface features of mouse incisors in the three groups. (**E**) Results of grading the degree of incisor damage in mice. (**F**) F^−^ concentrations in bone (femurs) of mice. Ctrl: Ctrl group; NaF: sodium fluoride group; NaF + Ba: co-administration of *Bifidobacterium* and sodium fluoride group. *** *p* < 0.001 indicates significant differences compared to the Ctrl group.

**Figure 2 foods-13-01011-f002:**
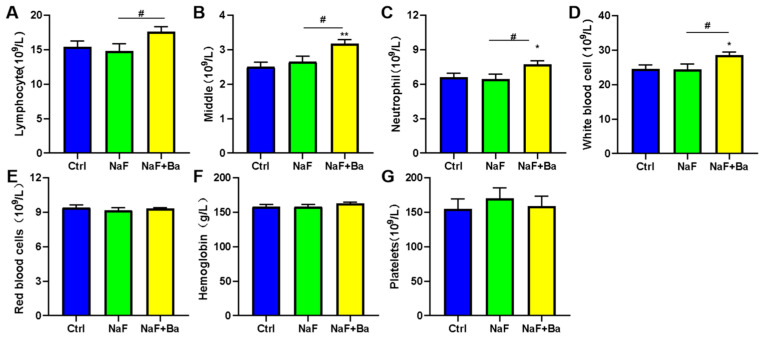
*Bifidobacterium* supplementation improves the alterations induced by fluoride in blood routine of mice. (**A**–**G**) Changes in blood routine of mice from Ctrl, NaF, and NaF + Ba groups, including the number of lymphocytes, middle cells (sum of monocytes, eosinophils, and basophils), neutrophils, white blood cell, red blood cells, hemoglobin concentration, and platelets. Ctrl: Ctrl group; NaF: sodium fluoride group; NaF + Ba: co-administration of *Bifidobacterium* and sodium fluoride group. * *p* < 0.05 and ** *p* < 0.01 indicate significant differences compared to the Ctrl group. # *p* < 0.05 points to significant differences compared to the NaF group.

**Figure 3 foods-13-01011-f003:**
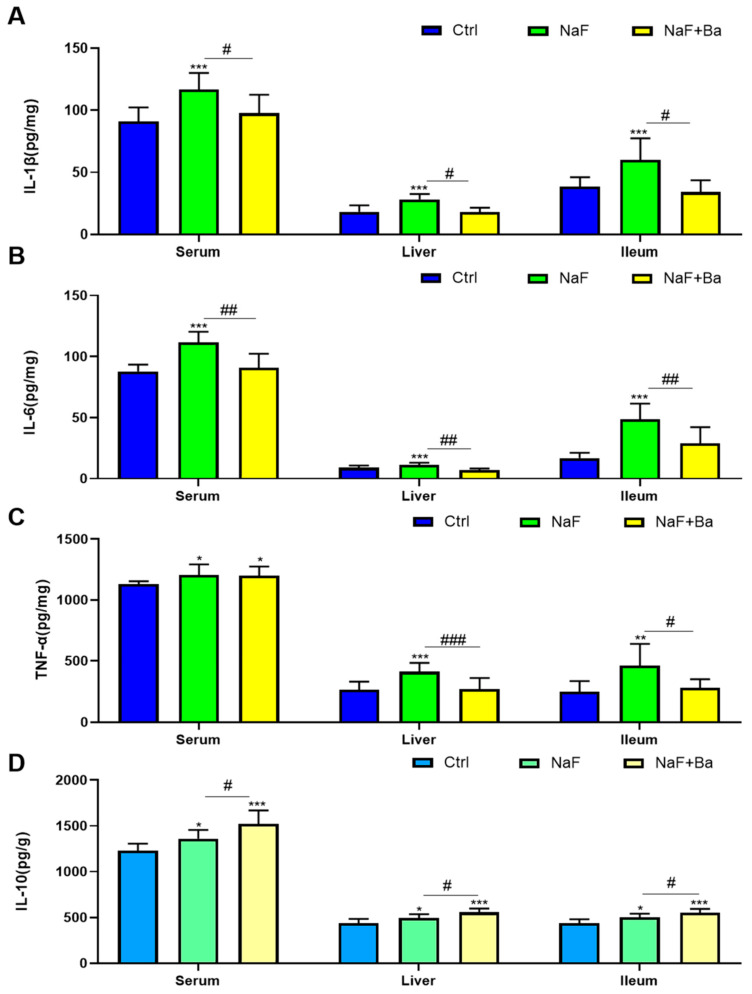
*Bifidobacterium* supplementation alleviates fluoride-induced alterations in inflammatory factors levels in serum, liver, and ileum of mice. (**A**–**C**) The histogram of changes in pro-inflammatory factor IL-1β, IL-6, and TNF-α contents in serum, liver, and ileum tissue of mice. (**D**) The levels of anti-inflammatory factor IL-10 in serum, liver, and ileum tissue of mice. Ctrl: Ctrl group; NaF: sodium fluoride group; NaF + Ba: co-administration of *Bifidobacterium* and sodium fluoride group. * *p* < 0.05, ** *p* < 0.01 and *** *p* < 0.001 indicate significant differences compared to the Ctrl group. # *p* < 0.05, ## *p* < 0.01 and ### *p* < 0.001 point to significant differences compared to the NaF group.

**Figure 4 foods-13-01011-f004:**
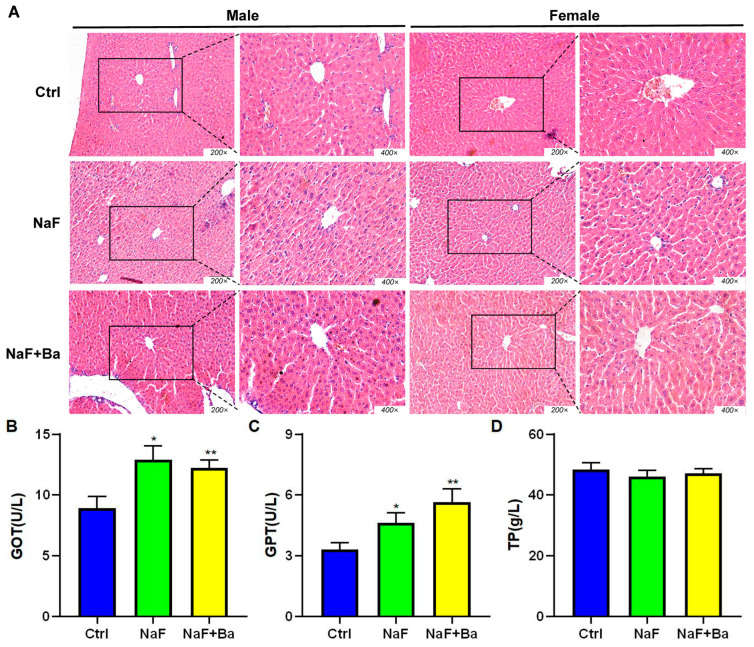
Fluoride induces liver morphology and function injury and *Bifidobacterium* intervention in mice. (**A**) The representative morphological images of liver tissue in mice (H.E staining). (**B**–**D**) Serum levels of the liver function marker enzymes aspartate aminotransferase (GOT), alanine aminotransferase (GPT), and total protein (TP) in mice. Ctrl: Ctrl group. NaF: sodium fluoride group; NaF + Ba: co-administration of *Bifidobacterium* and sodium fluoride group. * *p* < 0.05 and ** *p* < 0.01 indicate significant differences compared to the Ctrl group.

**Figure 5 foods-13-01011-f005:**
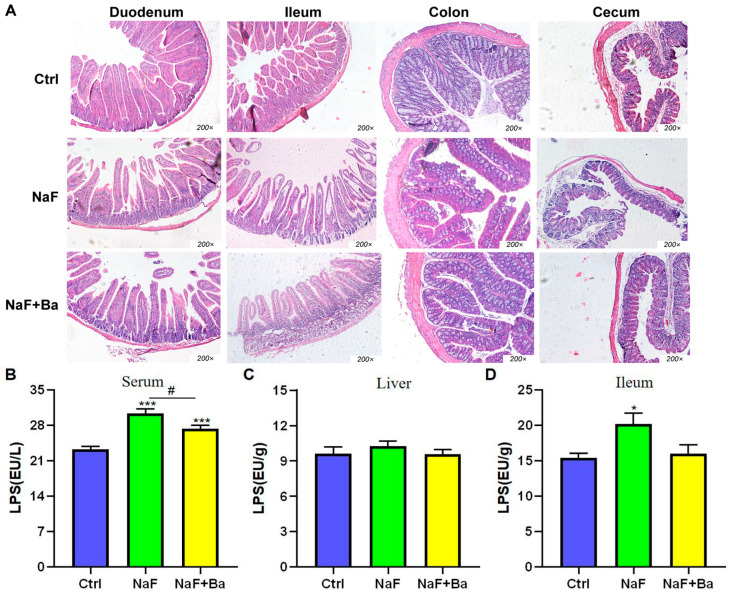
Fluoride induces the change of intestinal morphology and LPS content and *Bifidobacterium* intervention in mice. (**A**) The representative morphological images of the duodenum, ileum, colon, and cecum in mice (H.E staining, 200×). (**B**–**D**) The histogram of changes in LPS content in serum, liver, and ileum tissue of mice. Ctrl: Ctrl group; NaF: sodium fluoride group; NaF + Ba: co-administration of *Bifidobacterium* and sodium fluoride group. * *p* < 0.05 and *** *p* < 0.001 indicate significant differences compared to the Ctrl group. # *p* < 0.05 points to significant differences compared to the NaF group.

**Figure 6 foods-13-01011-f006:**
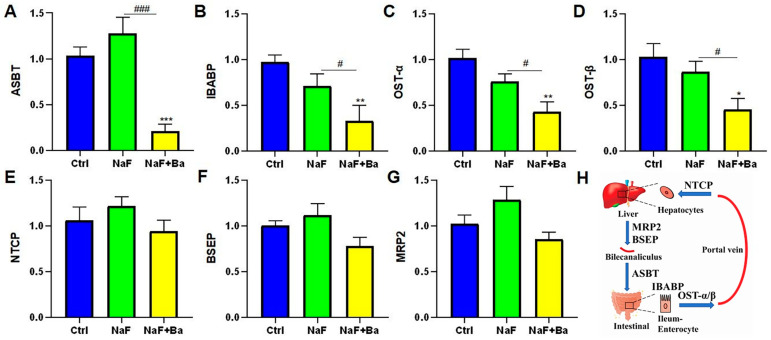
*Bifidobacterium* regulated fluoride-induced changes in bile acid transporters in liver and ileum. (**A**–**D**) The mRNA expression levels of bile acid transporter genes ASBT, IBABP, OST-α, and OST-β, in the ileum of mice. (**E**–**G**) The mRNA expression levels of bile acid transporter genes NTCP, BSEP, and MRP2 in the liver of mice. (**H**) Distribution of bile acid transporters in the hepatic–intestinal circulation. Ctrl: Ctrl group; NaF: sodium fluoride group; NaF + Ba: co-administration of *Bifidobacterium* and sodium fluoride group. * *p* < 0.05, ** *p* < 0.01 and *** *p* < 0.001 indicate significant differences compared to the Ctrl group. # *p* < 0.05 and ### *p* < 0.001 point to significant differences compared to the NaF group.

**Figure 7 foods-13-01011-f007:**
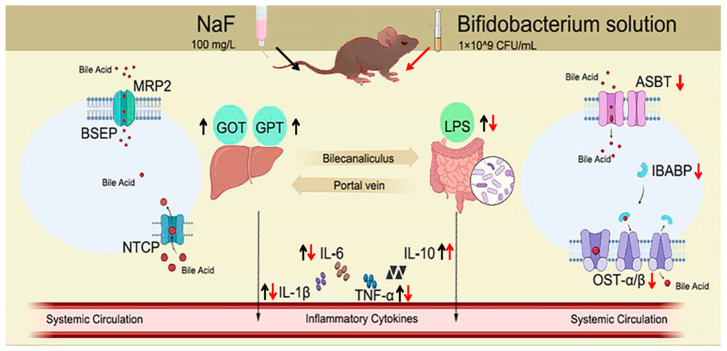
Schematic diagram of how *Bifidobacterium* mitigated fluoride-induced enterohepatic toxicity via inflammatory response and bile acid transporters (Material from Biorender).

**Table 1 foods-13-01011-t001:** qRT-PCR primer sequences.

Gene Name	Primer Sequences	Accession No.	Product Sizes (bp)
*β-actin*	F:TACCCAGGCATTGCTGACAG R:ACTCCTGCTTGCTGATCCAC	NM_007393.5	98
*NTCP*	F:ATGCCTTTCACTGGCTTCC R:CTGTTTCCATGCTGATGGTG	NM_011387.2	91
*BSEP*	F:GGACAATGATGTGCTTGTGG R:AGGGCCATTCTGAGATGTTG	NM_021022.3	80
*MRP2*	F:AAGCAGGTGTTCGTTGTGTG R:ACAGGAGGAATTGTGGCTTG	NM_013806.2	95
*ASBT*	F:CATGACCACTTGCTCCACAC R:AATCGTTCCCGAGTCAACC	NM_011388.3	91
*IBABP*	F:GTCTTCCAGGAGACGTGATTG R:CCGAAGTCTGGTGATAGTTGG	NM_008375.2	114
*OST-α*	F:TGCATCTGGGTGAACAGAAC R:AGCGATCTGCCCACTGTTAG	NM_145932.3	116
*OST-β*	F:GACCTGCATCTTGATGACTCC R:GGCCAAGTCTGGTTTCTCTG	NM_178933.2	90

**Table 2 foods-13-01011-t002:** Effect of *Bifidobacterium* on organ coefficient of fluoride-treated male mice (n = 6, mean ± SEM).

Male	Liver	Kidney	Spleen	Testis	Left Epididymis (10^−2^)	Brain
Crtl	3.53 ± 0.20	1.33 ± 0.05	0.21 ± 0.07	0.82 ± 0.03	17.52 ± 1.28	1.75 ± 0.10
NaF	3.56 ± 0.16	1.33 ± 0.04	0.24 ± 0.01	0.72 ± 0.10	15.37 ± 1.43	1.77 ± 0.12
NaF + Ba	3.59 ± 0.06	1.24 ± 0.04	0.25 ± 0.01	0.82 ± 0.03	17.44 ± 1.96	1.84 ± 0.05

**Table 3 foods-13-01011-t003:** Effect of *Bifidobacterium* on organ coefficient of fluoride-treated female mice (n = 6, mean ± SEM).

Female	Liver	Kidney	Spleen	Ovary (10^−2^)	Uterus	Brain
Crtl	3.55 ± 0.16	1.26 ± 0.05	0.27 ± 0.03	5.49 ± 0.82	0.23 ± 0.07	2.14 ± 0.12
NaF	3.60 ± 0.12	1.23 ± 0.04	0.30 ± 0.04	4.68 ± 1.62	0.26 ± 0.11	2.14 ± 0.11
NaF + Ba	3.41 ± 0.19	1.21 ± 0.04	0.25 ± 0.03	5.30 ± 1.12	0.23 ± 0.07	2.19 ± 0.07

## Data Availability

The original contributions presented in the study are included in the article, further inquiries can be directed to the corresponding author.

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
