# Peer review of "Bifidobacterium Relieved Fluoride-Induced Hepatic and Ileal Toxicity via Inflammatory Response and Bile Acid Transporters in Mice"

_foods, 2024, doi:10.3390/foods13071011_

Round 1

Reviewer 1 Report

Comments and Suggestions for Authors

The topic is of interest and in general, the experimental design is appropriate. However, there are several aspects that need to be addressed: 

- Lines 41-43: Please rewrite the sentence as it is confusing.

- Lines 46-47: Regarding the abundance of Bifidobacterium in the gut, please specify the host you are referring to.

- Lines 49-52: These statements are incorrect. Firstly, references 16 and 17 do not contain that kind of information. Secondly, what does "reducing the expression of LPS" actually mean? From who?

- Line 72: Please indicate the frequency and type of vehicle for oral administration of Bifidobacterium.

- Line 106: Please provide a justification for the determination of white blood cells, considering that the determination of neutrophils, lymphocytes, and middle cells were included.

- Lines 117-118: Please, describe the protocol for obtaining liver and ileum samples that were analyzed by ELISA.

- Lines 180-181: The presented studies do not provide enough evidence to make such a simplified suggestion. Further studies, such as analysis of activation in lymphocyte populations, may be needed.

- Lines 187-190: This sentence is extremely confusing. Please, clarify.

- Item 3.3: The visual differences between the NaF and NaF+Ba groups are not readily apparent in the images. In fact, the NaF+Ba group resembles the NaF group more than the control group. Additionally, the second image of the male control does not seem to correspond to a magnification of 400x. Please review the results and, if necessary, rewrite the subheading.

- Subheading 3.4 is unintelligible.

- Lines 246-248: The presented studies do not provide enough evidence to make such a categorical conclusion. 

- Lines 266-268: Regarding the expression of bile acid transporters, no changes were observed in the NaF group compared to the control group, indicating no fluoride-induced effects. So, rewrite the conclusion.

- Figure 7: The figure does not indicate the effects of NaF and Bifidobacterium administration.

- Discussion: Make any necessary changes after reviewing the results presented.

Comments on the Quality of English Language

I suggest a review by a native English speaker.

Reviewer 2 Report

Comments and Suggestions for Authors

Please respond to the suggestions and questions marked in the attached file.

Comments on the Quality of English Language

The wording of some sentences should be reviewed. Sentences that need to be revised and some writing suggestions have been marked in the attached file.

Round 2

Reviewer 1 Report

Comments and Suggestions for Authors

The manuscript was significantly improved. However, there are still some points that require revision and correction.

- Lines 111-112: This information was included in next paragrah.

- Subheading 3.2 should be rewriten (Suggestion: "Bifidobacterium alleviates fluoride-induced liver and ileal inflammation")

- Line 256: "... in the chroma of serum TNF-α..." should be replaced by "... in the level of serum TNF-α ..."

- Subheading 3.3: This statement is not consistent with what was said in both the results and the discussion section (lines 379-380).

- Figure 4: in the new version of the figure, the Y-axis labels and statistical references are absent.

- Line 341: In the studies presented, there was no evidence of interaction between Bifidobacterium and bile acid transporters. Please, correct the sentence.

- Lines 386-390: Since IL-10 behaves differently from the other cytokines, it should not be included in the same sentence. 

- Lines 391-393: The sentence requires improvement.

- Lines 395-396: Expressed in this way, the hypothesis is ambiguous as the results indicate that Bifidobacterium induces an anti-inflammatory response. Please, rewrite the text.

- Line 414: "intestinal" should be replaced by "intestine"

- Lines 417-420: Please rewrite the sentence as it is confusing and requires improvement.

- Lines 421-423: Please provide the bibliographic reference.

- Line 437: "in vitro"???

Comments on the Quality of English Language

Some sentences are confusing and require improvement.
